# Poor Response to Checkpoint Immunotherapy in Uveal Melanoma Highlights the Persistent Need for Innovative Regional Therapy Approaches to Manage Liver Metastases

**DOI:** 10.3390/cancers13143426

**Published:** 2021-07-08

**Authors:** Brett M. Szeligo, Abby D. Ivey, Brian A. Boone

**Affiliations:** 1Division of Surgical Oncology, Department of Surgery, West Virginia University, Morgantown, WV 26508, USA; brszeligo@mix.wvu.edu; 2Cancer Cell Biology, West Virginia University, Morgantown, WV 26508, USA; adharold@mix.wvu.edu; 3Department of Microbiology, Immunology and Cell Biology, West Virginia University, Morgantown, WV 26508, USA

**Keywords:** uveal melanoma, uveal melanoma, isolated hepatic perfusion, percutaneous hepatic perfusion, immunotherapy, melphalan

## Abstract

**Simple Summary:**

Uveal melanoma is a cancer that develops from melanocytes in the eye. This disease is extremely rare and has a strong predilection for liver metastasis, which has a poor long-term prognosis with overall survival of 6 months and 1 year mortality of 80%. Immunotherapy and checkpoint inhibitors have revolutionized treatment for cutaneous melanoma but have largely been proven to be ineffective in uveal melanoma. The poor response associated with systemic chemotherapy and immunotherapy approaches coupled with the fact that most metastatic disease is isolated to the liver suggests a persistent need for regional treatment approaches. Of regional therapies, hepatic perfusion continued to be associated with the best survival outcomes of up to 27 months. Additional therapies that target the unique biology of uveal melanoma are desperately needed. Until these treatments are developed and proven, isolated hepatic perfusion remains a viable treatment option.

**Abstract:**

Uveal melanoma is a cancer that develops from melanocytes in the posterior uveal tract. Metastatic uveal melanoma is an extremely rare disease that has a poor long-term prognosis, limited treatment options and a strong predilection for liver metastasis. Median overall survival has been reported to be 6 months and 1 year mortality of 80%. Traditional chemotherapy used in cutaneous melanoma is ineffective in uveal cases. Surgical resection and ablation is the preferred therapy for liver metastasis but is often not feasible due to extent of disease. In this review, we will explore treatment options for liver metastases from uveal melanoma, with a focus on isolated hepatic perfusion (IHP). IHP offers an aggressive regional therapy approach that can be used in bulky unresectable disease and allows high-dose chemotherapy with melphalan to be delivered directly to the liver without systemic effects. Long-term median overall survival has been reported to be as high as 27 months. We will also highlight the poor responses associated with checkpoint inhibitors, including an overview of the biological rationale driving this lack of immunotherapy effect for this disease. The persistent failure of traditional treatments and immunotherapy suggest an ongoing need for regional surgical approaches such as IHP in this disease.

## 1. Introduction

Uveal melanoma is an uncommon malignancy that develops from melanocytes in the posterior uveal tract of the eye. This disease is extremely rare, but has a strong predilection for liver metastasis, with 90% of uveal melanoma metastasizing to the liver and up to 50% of patients with uveal melanoma developing metastases [1]. Metastasis of uveal melanoma has a long latency period, often occurring many decades after treatment of the primary tumor with exclusive metastasis through hematogenous routes [2]. Primary tumor cells invade into systemic circulation and often metastasize to hemopoietic tissue and the liver [3]. Metastatic uveal melanoma has a poor prognosis, with median survival of 6 months and 1 year mortality of 80% [1,4]. Surgical resection is rarely possible due to extent of disease. Because there are limited systemic treatment options and metastases are almost exclusively isolated to the liver, an aggressive regional approach has been pursued. Isolated hepatic perfusion (IHP), which involves perfusion with melphalan through a liver circuit with venovenous bypass, has been studied for uveal melanoma. IHP allows for high concentrations of more effective chemotherapy (melphalan) to be delivered to the liver without systemic toxicity. Survival outcomes for metastatic uveal melanoma can be extended with IHP treatment, achieving a median overall survival (OS) of up to 24 months [5]. However, IHP is an aggressive surgical approach, with significant morbidity and risk of mortality. Recently, systemic immunotherapy has revolutionized the management of cutaneous melanoma and many providers attempt to extrapolate these findings to uveal melanoma. The low mutational rate in uveal melanoma compared to cutaneous melanoma contributes to the failure of immunotherapy [6]. Mutations and chromosomal abnormalities contribute to the development of uveal melanoma while different gene mutations determine propensity of metastasis. We sought to review the existing literature regarding the efficacy and biological basis for treatment with immunotherapy in uveal melanoma, recent development of targeted agents for this disease, as well as discuss the history and outcomes related to regional approaches such as IHP.

## 2. Primary Treatment for Localized Uveal Melanoma

Primary uveal melanoma can be treated locally depending on the size and location of the tumor, comorbidities of the patient, and patient preference. Traditionally, enucleation of the eye was the treatment of choice but this significantly impacts patient quality of life, resulting in exploration of better treatment options. This led to advances in chemotherapies and radiation techniques that are as effective as enucleation and provide the patient a better quality of life post-treatment [7]. Primary resection of the tumor can be trans-scleral (exoresection) or trans-retinal (endoresection). Exoresection involves creating a scleral flap to resect of the tumor, while sparing the normal retina and vitreous [8]. This technique preserves normal retina, which increases visual preservation [8]. Exoresection is a difficult procedure for tumors near the optic disc and can lead to incomplete resection and reoccurrence [9]. Endoresection involves tumor resection after 3-port pars plana vitrectomy with retinal flap [8]. This procedure allows for better resection of posterior tract tumors [9]. Both exoresection and endoresection can be combined with radiotherapy to prevent tumor recurrence.

Brachytherapy has become the most common treatment for primary uveal melanoma. A radioactive plaque is sutured to the episcleral layer to deliver a fixed dose directly to the tumor. This type of treatment does come with limitations and requires regular examinations to look for radiation induced damage for 2 to 5 years following treatment. Another option is charged particle radiation therapy, which has been proven more effective for larger tumors not suitable for brachytherapy. Teleradiotherapy with proton beam and fractionated stereotactic irradiation is a conservative option in localized tumors that provide high-dose radiation in a controlled location [10]. Proton beam and stereotactic irradiation therapy are often eye preserving, maintains vision in >50% of patients and provides excellent local control, especially in tumors located near the macula or optic disc [10]. Stereotactic irradiation provides local radiation using Gamma Knife which delivers radiation from several directions to the tumor. The downside of local radiation is risk of increased intraocular pressure and optic neuropathy [10]. Additional options include photocoagulation, transpupillary thermal therapy, and photodynamic therapy if the prior are not suitable or successful [11]. Adjuvant therapy has also been utilized as a treatment option in conjunction with the radiation techniques above to reduce the risk of metastasis.

The standard of care for treatment of primary tumor remains undefined. However, novel globe-sparing approaches have provided options other than traditional enucleation. Enucleation is reserved for large primary tumors, while globe-sparing radiation with plaque brachytherapy or teletherapy can be used for smaller tumors [12]. While novel therapies have been developed for the primary tumor treatment of uveal melanoma, the OS and rate of metastasis remain unchanged [13]. Once metastasis has occurred, aggressive primary tumor treatment is no longer warranted [14].

## 3. Traditional Treatment Options for Metastatic Uveal Melanoma

Uveal melanoma can spread hematogenously to other regions of the body and metastasize in other organs. In cases of uveal melanoma, the 5 and 10 year rates of metastasis are 25% and 34%, respectively, with up to 50% of patients developing metastases at some point in disease course [1]. In approximately 90% of cases, there is a predilection for metastasis to the liver [1]. There are various methods of treatment for isolated liver metastases. There have been limited clinical trials for uveal melanoma with liver metastasis, with many of these trials failing to demonstrate benefit due to lack of an effective response.

### 3.1. Systemic Therapy

Traditional systemic chemotherapies, a mainstay for most disseminated tumors, are largely ineffective in uveal melanoma cases [15]. Early experimental studies using nitrogen mustard compounds for treatment of metastatic disease from uveal melanoma initially proved to be an effective treatment with most tumors showing a therapeutic response [16]. However, treatment was limited by toxicity. The most commonly used systemic therapies include alkylating agents (nitrogen mustard compounds, dacarbazine, temozolomide, nitrosoureas, cisplatin) and microtubule inhibitors (vinca alkaloids).

Bedikian et al. [17] studied the effects of temozolomide, an alkylating agent similar to dacarbazine, that is activated in the GI tract, which allows a high concentration to enter the liver on first pass [17]. Patients received temozolomide every day for 21 days for 4 weeks. There were no responders and two patients with stable disease. The median OS was 6.7 months and the progression-free survival (PFS) was 1.84 months. The authors concluded temozolomide was an ineffective treatment in metastatic cases [17].

Pyrhönen et al. [18] looked at the response in 20 patients who received bleomycin, dacarbazine, and lomustine [18]. No patients had a complete response (CR), only 15% showed a partial response (PR) and 55% had stable disease (SD). The median PFS was 4.4 months and the median OS was 12.3 months. Complications of therapy included 40% grade 3–4 hematologic toxicity with leukopenia or thrombocytopenia. Two patients had treatment-related death.

Another study looked at the synergetic effects of treosulfan and gemcitabine (an antimetabolite drug) [19]. Patients received gemcitabine plus treosulfan (*n* = 24) or treosulfan alone (*n* = 24). While the treosulfan plus gemcitabine showed slightly more responses compared to treosulfan alone, the overall response was extremely poor. The PFS was 3 months in the combination treatment group and 2 months in the treosulfan alone cohort.

Layvraz et al. [20] reviewed systemic chemotherapy agents, alone and in combination with multiple agents, or combined with immunotherapy. The response rate between all groups studied was 0–15%, with a median OS of 6–12 months [20]. Another study by Pons et al. [21] retrospectively reviewed outcomes in patients with metastatic uveal melanoma. Twenty-five patients received systemic chemotherapy, and 23 patients were treatedwith best supportive care [21]. Chemotherapy consisted of dacarbacine (*n* = 13), temozolamide with interferon (*n* = 4) or without interferon (*n* = 1), fotomustine (*n* = 5), and carboplatin/dacarbacine/interferon-a/interleukin-2 (*n* = 2). The OS in all groups was 10.83 months, the OS in the chemotherapy group was 10.83 months and the OS in the supportive care group was 8.03 months [21]. The numerous systemic chemotherapies reviewed had a median OS ranging from 6 to 12 months [21]. The authors conclude that patients should consider newer novel therapies with greater efficacy then traditional chemotherapy.

### 3.2. Surgical Resection and Local Ablation

Surgical resection is the preferred treatment for patients who are medically fit for surgery with liver metastases that are resectable; although this is rarely possible in patients with uveal melanoma, due to the diffuse pattern of disease [22]. Hepatic metastasectomy in combination with systemic therapy has demonstrated the best long-term survival results [15]. Aubin et al. [23] reported a meta-analysis of twenty-two studies examining 579 patients who underwent liver resection. The median OS ranged from 14 to 41 months for surgical resection, while non-surgical treatment resulted in a median OS range of 4–12 months [23]. In cases where disease is fairly localized but surgical resection is not feasible, percutaneous, laparoscopic or open ablation may be considered. Ablation techniques utilize electrical current or microwave to cause tissue destruction and is best used for localized, small tumors (< 3 cm) and cannot be used in extensive widespread disease [24].

### 3.3. Regional Therapies (TACE, Y90, Hepatic Artery Infusion Chemotherapy)

Regional therapy approaches are best utilized to treat the entire liver parenchyma in the setting of diffuse disease. Hepatic artery infusion (HAI) and transarterial chemoembolization (TACE) deliver chemotherapy directly to the liver through arterial circulation [25]. While cytotoxic drugs that have efficacy in uveal melanoma, such as melphalan, can be delivered via these routes, the drug does reach the systemic circulation, limiting the dose used and resulting in systemic toxicity [16].

HAI allows for arterial delivered chemotherapy directly to the liver. Liver metastases receive blood supply from hepatic artery, while normal liver receives the majority of blood from portal vein [26]. This allows for directed infusion of high-dose chemotherapy directly to the liver metastases, sparing normal hepatocytes from the full effect of treatment. This approach favors selected chemotherapy agents that are hepatically cleared to prevent chemotherapy reaching systemic circulation, causing systemic side effects. Most commonly used chemotherapies are floxuridine, oxaliplatin, irinotecan, 5-fluorouracil, and mitomycin C. We have previously reported a series of 14 patients that could not undergo IHP due to hepatic dysfunction and underwent HAI of melphalan. The 30-day mortality was 21%, the median OS was 2.9 months and 21% of patients achieved nearly 1 year survival [27]. While these patients were very high risk for treatment, they may have shown some benefit to therapy. HAI might be better utilized as an adjuvant therapy following more aggressive primary therapy with IHP.

Benefits of chemoembolization (TACE) include the directed delivery of chemotherapy to the liver and embolization of arterial supply to the tumor, resulting in necrosis. Advances in drug-eluting beads allow prolonged sustained delivery of chemotherapy and embolization. However, extensive liver involvement is a contraindication to this therapy. One study evaluated the effect of drug-eluting beads with irinotecan (DEBIRI) and intravenous dacarbazine (DTIC) [28]. The OS for DEBIRI was 9.4 months compared to 4.6 months in DTIC group [28]. CT imaging at first follow up showed no patients with complete or partial response, with 11 of 13 patients having progressive disease [28]. The authors of this study concluded that DEBIRI provided no benefit compared to other chemoembolization therapies [28].

Another study by Sharma et al. [29] looked at hepatic chemoembolization using cisplatin, doxorubicin and mitomycin C. Twenty patients received a mean of 2.4 chemoembolization treatments. The results showed no mortality within 30 days, with a median OS of over 10 months. A total of 65% of patients had SD, with 35% having disease progression [29]. The authors concluded that chemoembolization is more effective in nodular angiographic pattern compared to infiltrative angiographic pattern, with a median OS of 25 and 3.6 months, respectively [29]. In 1995, Bedikian et al. [30] reviewed HA chemoembolization with cisplatin, HAI, and systemic therapy, concluding HA chemoembolization had the best outcomes with median survival of 14.5 months [30].

Radioembolization with Y90 is a similar method to TACE, replacing chemotherapy with radiotherapy. Beads filled with isotope yttrium Y90 are directed to the liver metastases. Arterial supply to the tumor is embolized and the bead releases high-dose radiation directed at the site of the tumor. With radiotherapy, there is less concern for systemic leak of toxic chemotherapy. A study by Klingenstein et al. [31] looked at 13 patients that received radioembolization [31]. A total of 10 patients had previously received chemotherapy with dacarbazine and cisplatin and showed no response. After radioembolization, the median OS was 7 months. This poor response questioned the utility of radioembolization treatment.

## 4. Rationale for Isolated Hepatic Perfusion in Uveal Melanoma

Nitrogen mustard compounds (melphalan) belong to the alkylating antineoplastic class of drugs. They work by inhibiting DNA synthesis and division, leading to cell death. These compounds proved to be an effective treatment for most uveal melanoma metastasis [16]. However, the dose and concentration needed to achieve this response were limited due to cytotoxic effects, mainly bone marrow suppression. This resulted in limited use and tumor recurrence [16]. Development of a variety of delivery methods to avoid systemic cytotoxic effects followed. In 1958, Creech et al. [16] developed an isolated liver circuit in a canine model that used up to 2.0 mg/kg HN_2_, nitrogen mustard diazenide compound, without systemic toxicity [16]. Ausman first preformed IHP in humans in 1960, demonstrating a potential treatment option [32]. However, due to the aggressive nature of the procedure and technical complexity in the setting of uncertain outcomes, this procedure was not utilized again until the 1990s.

## 5. Surgical Approaches and Treatment Outcomes for Isolated Hepatic Perfusion

IHP offers an aggressive regionalized approach for treatment of metastatic liver disease. IHP is often used for bulky unresectable liver lesions with extensive liver involvement, requiring the need for an aggressive approach. High concentrations of drug can be delivered to the liver tumor without exposure to systemic circulation, that would otherwise be cytotoxic to the patient. Patients without signs of extrahepatic disease are potential candidates for IHP.

IHP is a technically demanding and extensive procedure. The procedure first begins with an exploratory laparoscopy or laparotomy to confirm no evidence of extrahepatic disease. The liver is mobilized and the inferior vena cava (IVC), portal triad and gastroduodenal arteries are dissected. A prophylactic cholecystectomy is performed. The isolated liver circuit consists of two separate components, a venovenous bypass and a perfusion circuit (Figure 1). Venovenous bypass recirculates deoxygenated blood from lower extremities back to the heart, bypassing the vena cava. The venovenous bypass circuit is established by insertion of a cannula into left femoral vein through the infrarenal IVC and a second cannula from the internal jugular (IJ) into superior vena cava (SVC). The perfusion circuit consists of an inflow and outflow cannula. An inflow cannula is inserted through an arteriotomy in the gastroduodenal artery (GDA) and secured in proximal GDA. A cross clamp is placed on the portal hepatis (common hepatic artery, bile duct and portal vein) to occlude arterial flow to the liver. The outflow cannula is inserted from the femoral vein into the retrohepatic vena cava. The vena cava is clamped superiorly at the level of suprahepatic IVC and inferiorly at the level of suprarenal infrahepatic IVC. The perfusion circuit is complete and recirculates perfusate to the liver without reaching systemic circulation. Perfusate is circulated at 600–1200 mL/min and heated to 40 °C for 1 h. The liver circuit is washed with 3L to remove any remaining drug before liver flow is reestablished.

Open IHP is a treatment option that has been implemented for unresectable diffuse liver metastasis. Table 1 shows review of the literature of open IHP for uveal melanoma. The median OS is 15.3 months (range 9.9–24 months). Most cases used melphalan as perfusate. Some studies investigated the use of tumor necrosis factor alpha (TNFα). Alexander et al. [33] had a total of 22 patients receiving melphalan (*n* = 11) and melphalan and TNFα (*n* = 11). The OS was 11 months. Previous studies suggest that TNFα prolongs the overall duration of response to treatment [33]. Ben-Shabat et al. [34] reported on 68 total patients, with only 2 patients receiving melphalan and TNFα. The median OS was 22.4 months. Due to the limited number of patients receiving TNFα, the effect of TNFα on prolonging response and the median OS cannot be determined [34]. The most responsive perfusate/combination has yet to be determined, requiring the need for more review of response to perfusate and whether combinations of drugs offer any advantage. However, open IHP with melphalan alone appears to be a successful treatment method. Olofsson et al. [5] reported 34 total patients treated with melphalan IHP. The median OS was compared to patients that survived the longest without IHP treatment. Patients treated with open IHP had 14 month extended survival compared to patients not receiving IHP [5]. However, IHP is not always successful. Varghese et al. [35] had a 50% success rate, 8 patients showing complete (*n* = 1) or partial (*n* = 7) response and 8 patients showing no response [35]. The median OS regardless of perfusate suggests that this IHP does offer a promising treatment method, potentially extending survival for patients with metastatic uveal melanoma.

IHP can be associated with significant morbidity. Patients undergo laparotomy and extensive dissection as part of the procedure and can suffer from any of these common complications associated with major abdominal surgery, including wound infection, intra-abdominal abscess formation, and bleeding complications. There is the potential for cytopenia and bone marrow suppression from systemic leak of chemotherapy. Leak monitoring can be performed using radiolabeled serum albumin and monitoring counts using a gamma counter. However, given the relative low risk of systemic leak, this is rarely performed. Instead, volumes in the circuit are closely monitored for evidence of loss that would suggest a leak. Advanced inflow and outflow circuits have resulted in near 0% leak rate at some institutions. Other morbidity is patient dependent and depends on serval factors including amount of pre-IHP chemotherapy given, liver function, presence of cirrhosis or portal hypertension, and age >70 [22]. Transient grade 3 or 4 hepatotoxicity can occur in 40–70% of cases [22]. Tumor involvement can affect mortality, with up to 25% mortality in cases with > 50% tumor involvement of the liver [22]. In a review of 90 cases of IHP, results showed 3% perioperative mortality and 5.5% irreversible liver failure [22]. Patients typically have inpatient recovery with median hospital stay of 11–13 days and operative time of approximately 8–9 h. Ben Shabat et al.’s [34] study of 68 patients showed 7% mortality within 30-day post-procedure due to hepatic failure [34]. There was also 9.5% Clavien-Dindo grade 3 or 4 complications, including 2 patients with respiratory insufficiency, 1 patient with liver failure with necrosis, and 1 patient requiring reoperation due to bleeding [34].

While much of the early literature has focused on an open technique IHP, a percutaneous minimally invasive approach is also possible and gaining traction. This approach offers several potential benefits of faster recovery, shorter hospital admission and the ability to perform the procedure multiple times. Due to the nature of the extensive open technique, it is not possible to reopen the abdomen multiple times to perfuse the liver. Although percutaneous hepatic perfusion (PHP) offers a less invasive method it does come with risks. Due to the minimally invasive nature of this procedure, it is associated with a greater rate of systemic exposure and is also difficult to confirm lack of extrahepatic disease. A double balloon catheter occludes flow, creating and isolated outflow circuit. The first balloon is inserted in the IVC superior to the hepatic veins while a second balloon is inserted in the IVC caudal to the liver. Using angiographic guidance, the hepatic artery is accessed percutaneously from the femoral artery. Perfusate is delivered through the hepatic artery and returned through hepatic veins and the outflow circuit. Unlike the open procedure, perfusate blood is filtered to remove drug and returned to systemic circulation through a cannula inserted into the internal jugular vein.

In contrast to the major abdominal surgery involved with IHP, PHP is a shorter procedure, lasting approximately 3–4 h with a hospital stay of only 2–3 days. As such, PHP has the potential to be associated with less surgical morbidity. Despite this potential lower complication risk, the lack of randomized studies and limited number of case series for PHP, preclude definitive comparisons. In a study by Karydis et al. [44], 51 patients underwent PHP with no treatment-related deaths [44]. A total of 37.5% of patients had grade 3–4 non-hematologic toxicity, 19.6% with perioperative bleeding, 13.7% with thromboembolic events, 27.4% with grade 3–4 thrombocytopenia and 31.4% with grade 3–4 anemia, 43.1% with late neutropenia, and 29.4% with mild and transient transaminitis [44]. The increase in adverse events of PHP contribute to a greater risk of systemic leak of cytotoxic chemotherapy.

Table 2 shows a review of the literature of PHP for metastatic uveal melanoma. The median OS using melphalan is 17.3 months (range 9.6–27.4 months). A study of 93 patients comparing percutaneous hepatic infusion of melphalan (*n* = 44) compared to the best-available care (BAC, *n* = 49), including systemic chemotherapy, embolization and supportive care endorsed PHP with melphalan as a primary treatment option [45]. The median overall PFS in the PHP-Mel group was 5.4 months and in the BAC group, 1.6 months [45]. The OS was not sufficiently different between the two groups (10.6 month in PHP-Mel group and 10.0 months in BAC group) [45]. There was a significant difference in the overall response rate between the two groups, 27.3% in PHP-Mel group and 4.1% in BAC group, with all responses being PR [45]. The authors note that 57.1% of patients in the BAC group were able to crossover, confounding the OS endpoint [45]. However, the tumor response rate of PHP-Mel compared to BAC mainly consists of ineffective therapy and methods, suggesting that hepatic perfusion with melphalan is a viable option.

One study achieved a median OS of 27.4 months, which is longer than previous reports of a maximum median OS of 24 months [5,46]. This is a significant prolongation of survival compared to no treatment, which has an average survival of only 6 months. An advantage of PHP as discussed previously is that the procedure can be performed multiple times. Vogl et al. [47] had several patients receiving multiple treatments, with two patients receiving this procedure four times [47]. These two patients did not respond well to treatment and ultimately had progressive disease. However, there were patients that had more than one procedure that showed a response to repeated treatments.

Limitations to IHP are focused on the extent of the technically challenging procedure. Patient selection is important. IHP is not indicated in cases where the patient has evidence of extrahepatic disease and must have unresectable liver lesions. Patient characteristics include amount of pre-operative chemotherapy given, liver function (bilirubin >1.5 mg/dL, elevated INR, platelet <100,000), presence of cirrhosis or portal hypertension and age >70 are relative contraindications to surgery [22]. Patients must also be good surgical candidates and be able to withstand the extensive surgery and recovery time. Due to extensive/aggressive nature and associated high morbidity of open IHP, the innovation of PHP offers potential for quicker recovery and fewer limitations. A major difference in limitations between the two procedures is the ability to perform PHP multiple times where open IHP can only be done once. However, PHP is associated with significant risk of intraoperative leak which can lead to adverse events and associated morbidity/mortality. Future development of more effective minimally invasive approaches are warranted.

## 6. Exploration into Immunotherapy for Treatment of Metastatic Uveal Melanoma

Immunotherapy with checkpoint inhibitors has revolutionized the care of cutaneous melanoma, but these drugs are largely ineffective for uveal melanoma, with minimal therapeutic response [22]. The difference between treatment responses for cutaneous and uveal cases can be explained by critical differences in the cell biology between these two malignancies. Cutaneous and uveal melanoma have different rates of mutation. Uveal melanoma has low rates of mutation compared to cutaneous which has high rates [49]. Tumors with an increased rate of mutation have increased neoantigens which increases the chance of immune cell recognition, suggesting why cutaneous has better response to immunotherapy compared to uveal melanoma [49]. Joshua et al. [50] treated 11 patients in a phase 2 clinical trial with tremelimumab, an anti-CTLA4 immunotherapy, and showed no response to treatment [50]. Heppt et al. [51] reported that patients treated with ipilimumab (anti-CTLA-4) and either pembrolizumab or nivolumab (anti-PD-1) had a response rate of 16.7% and 33.3%, respectively, with only two patients experiencing adverse effects. Another group looked retrospectively at anti-PD-1 inhibition with and without ipilimumab to assess whether combination immunotherapy would lead to better treatment responses. Their group of patients that only received the PD-1 inhibitor showed an overall response rate of 4.7% with a poor median PFS and OS. These results have also been shown in other studies of similar characteristics [51,52,53]. Despite the limited success of these retrospective studies, prospective trials studying combined immunotherapy for uveal melanoma are currently underway (NCT01585194, NCT02626962) [51].

Dendritic cells are another form of immunotherapy that result in anti-tumor activity by stimulating T cells to destroy the tumor cells [49]. In a study by Bol et al. [54], 14 patients were treated 3 times with dendritic cell vaccination with autologous dendritic cells, antigen-presenting cells treated with melanoma antigen gp100 and tyrosinase [54]. A total of 29% of patients had an immune system response, with 10 patients having SD at 3 months post-vaccination and 7 patients having PD at 6 months [54]. The median OS was 19.2 months [54]. While the median OS with dendritic cell vaccination shows benefits, it had limited utility in cases of high tumor burden [54]. This treatment would not be feasible in metastatic uveal melanoma cases with a high burden of liver involvement. The authors further conclude that dendritic cell vaccination may be better utilized as an adjuvant therapy after primary treatment of uveal melanoma with no evidence of metastasis [54].

A recent study by Levey et al. [55] looked at the concurrent use of immunotherapy and Y90 [55]. A total of 24 patients were enrolled and 12 patients received Y90 followed by standard of care and 12 patients received Y90 with immunotherapy 3 months before or after radiotherapy. Immunotherapy consisted of ipilimumab (*n* = 8), nivolumab (*n* = 4), IL-2 (*n* = 4), and pembrolizumab (*n* = 4). Y90 with immunotherapy has an OS of 26 months compared to 9.5 months for Y90 alone. Y90 with immunotherapy, had no CR, 8% PR, 50% SD, and 42% PD. Y90 alone had no CR, 10% PR, 50% SD, and 40% PD. Patients receiving concurrent therapy had an overall PFS of 10.3 months compared to 2.7 months for Y90 alone. While concurrent Y90 and immunotherapy may show a synergetic effect, the OS was decreased in cases where tumor was >7 cm.

Newer research has focus on tebentafusp, which is an immune mobilizing monoclonal T-cell receptor against tumor cells. These molecules activate T cells to kill tumor-specific cells [49]. In an initial study of 14 patients with uveal melanoma treated with tebentafusp, 0% had CR, 14% PR, 57% SD and 29% PD [49]. All patients treated with tebentafusp experienced adverse events following treatment [49]. Currently, tebentafusp is only indicated in patients that are positive for HLA-0201, which is only seen in ~50% of Caucasians [49]. Long-term effectiveness and limitations of use need to be determined before this is an accepted treatment option.

The potential for immunotherapy as a treatment of uveal melanoma remains uncertain and requires more comprehensive analysis of how uveal cancer evades the immune system to determine better immunotherapy treatment options. Schank et al. [49] proposes several reasons why immunotherapy is ineffective in uveal melanoma: liver as primary site of metastases which responds poorly to immunotherapy, liver metastases tend to have high tumor burden, a low rate of mutation, and low CD8+ T cells within tumors [49]. Until then, immunotherapies appear unproven and ineffective for treatment of uveal melanoma.

## 7. Biological Basis for Failure of Immunotherapy in Uveal Melanoma

Immunotherapy has revolutionized the care of patients with cutaneous melanoma and dramatically improved outcomes [56]. Unfortunately, these drugs have not demonstrated similar outcomes for patients with uveal melanoma, with little to no response to immunotherapy treatment for this disease. Recently, there has been more understanding of the mechanisms and tumor biology behind this type of cancer that may provide insight as to why immunotherapies are not as effective.

Uveal melanoma is driven by different mutations than cutaneous melanoma, with a lower mutational burden in the tumor cells [57]. A higher mutation rate is associated with the relative effectiveness of immunotherapies such as anti-CTLA4/PD-1. This is thought to be due to the creation of neoantigens from the mutations that aid in generating a stronger immune response [53]. Because uveal melanoma has a lower mutation burden, it expresses fewer neoantigens, thereby generating less of an immune response when exposed to immune cells.

The eye is an immune-privileged organ, with less access to the immune system and contained in an anti-inflammatory environment [58]. This effectively prevents any immune-mediated damage that the organ might experience due to an immune response. However, it also limits an immune response to cancer. The eye has its own physical barrier to the immune system due to the distribution of blood to the different sections of the eye and the blood–uveal barrier. The blood–uveal barrier is generated by the tight junctions between the pigmented epithelial cells, endothelial cells of the retina capillaries, and the avascular cornea [59]. This not only helps prevent certain immune cells reaching different parts of the eye but also allows for the eye to create its own environment in the aqueous humor to prevent any activation of immune cells that might have gotten through the barrier [60].

Within the aqueous humor, the eye generates many different proteins and molecules that sustain an anti-inflammatory environment. NK cells are very important in in vitro experiments for cell lysis of uveal melanoma tumor cells, but this promising line of investigation has failed to be duplicated in vivo. This is likely due to the fact that the eye produces both TGF-Beta and MIF in the aqueous humor that inhibit the action of NK cells once they reach the tumor cells [61]. These properties also follow uveal melanoma cells as they metastasize to prevent NK-mediated cell death once they leave the eye [62]. Uveal melanoma cells also overexpress the MHC class 1 genes, further preventing NK cell-mediated tumor cell death [63]. Verbik et al. [64] discovered in 1997 that uveal melanoma causes a decrease in lymphocyte proliferation and hypothesized that the anti-inflammatory substances found in the aqueous humor were critical factors contributing to this [64].

Uveal melanoma cells have also developed a mechanism to alter their response to inflammation during the adaptive immune response. In the presence of IFN-gamma, cells generate a more anti-inflammatory response. IDO and PD-L1 are two of the genes expressed due IFN-gamma. Excess IDO works to dampen the T-cell response by inducing starvation through degradation of tryptophan, an essential amino acid for DNA replication [65]. The excess PD-L1 causes inactivation of any T cells that enter into the aqueous humor, preventing further immune activation against the tumor cells [66]. All three classes of complement regulatory proteins are upregulated in the eye (CD46, CD55, CD59), representing another type of anti-inflammatory molecule. This prevents any complement activation and protects the uveal melanoma tumor cells against complement-mediated cell lysis [67]. Cells in the eye also generate alpha-MSH, VIP, somatostatin, ascorbate and CGRP, which have been found to create a more anti-inflammatory environment, protecting the uveal melanoma cells from immune recognition [4,60,68,69,70].

Uveal melanoma tumor cells have adapted and used their environment in the eye to evade immune recognition. This immune privilege, along with the mechanisms behind carcinogenesis of uveal melanoma, has resulted in a roadblock for immunotherapies, making these drugs an ineffective treatment option for now. More information is being generated about this type of cancer and how it interacts with the immune system that will hopefully provide further insights into why current immunotherapies are not successful and unlock other immunotherapy strategies that will be beneficial.

## 8. Advances in Molecular Targeted Therapies in Uveal Melanoma

While traditional treatment options have not shown much benefit for patient survival with metastatic disease, molecular targets have proven to be the most promising of newer treatment modalities. The main driving mutation found in most uveal melanomas is in the GNAQ/GNA11 pathway. Activating mutations in this gene leads to activation of its downstream targets such as PLCB, PKC, and ERK1/2, all of which have been proven to be good drug targets [69,71]. However, these G protein mutations have no correlation to the risk of metastasis and occur in both metastatic and non-metastatic uveal melanoma [6]. *GNA11* mutations are seen in ~57% of metastasis [6]. The drug Cabozantinib-s-malate, a known inhibitor on MET and VEGF2, was tested in a phase 2 clinical trial with advanced metastatic diseases and showed an OS rate of 11.5 months in hepatocellular carcinoma (NCT00940225) [72]. It is now being used in another clinical trial to compare its efficacy compared to temozolomide plus dacarbazine, traditional treatment for metastatic uveal melanoma, (NCT01835145). MEK inhibition is also being assessed using the drug selumetinib with promising outcomes thus far [73]. Other targets, such as the growth factor receptor c-Kit was proven to be a potential therapeutic target for treatment of uveal melanoma in vitro. This lead to sunitinib being assessed in a clinical trial for its efficacy against uveal melanoma and was found to have no benefit on the PFS or OS [74]. HDAC inhibitors have been recently investigated as a new therapy for uveal melanoma. Mutations in *BAP1, SF3B1* and *EIFAX* have been shown in cases of metastasis. BAP1 is a deubiquinating enzyme encoded on chromosome 3 that acts as a tumor suppressor by regulated cell growth, proliferation, cell death, repair of DNA, and control of gene activity. *BAP1* mutations have high risk of metastasis and occur in ~50% of uveal melanoma and typically result in metastasis within 5 years [75]. *SF3B1* has intermediate risk and results in metastasis within 15 years and *EIF1AX* are low risk and rarely result in metastasis [75]. Chromosome 3 is often lost in uveal melanoma and its loss is highly associated with metastasis. HDAC inhibitors have been found to help reverse the effect of losing BAP-1 in uveal melanoma in vitro and shown to decrease metastasis rates [76,77]. Gain of chromosome 6p and 8q is also involved in mutations and risk of metastasis [75]. More evaluation of these drugs is needed to determine their full potential in treatment of uveal melanoma [78]. Treatment with small-molecule inhibitors is a very promising new treatment option for uveal melanoma and should be further evaluated to understand their full potential.

In a phase II trial, the synergetic effects of bevacizumab, a VEGF inhibitor, and temozolomide, an alkylating agent, were studied, assessing tumor angiogenesis. This study contained 36 patients with metastatic uveal melanoma and consisted of 6, 28 day cycles of bevacizumab (on day 8 and 22) and temozolomide (days 1–7, 15–21) [79]. Patients underwent a median of four cycles of therapy. The median PFS was 12 weeks and the OS was 10 months. No patients had a treatment response, 8 had SD and 27 progressed. After continued maintenance therapy with bevacizumab, 5 patients had SD ranging from 11 to 35 months. Piperno-Neumann et al. [79] showed through CT that uveal melanoma liver metastasis have increased perfusion compared to normal liver [79]. While blood flow and blood volume were reduced at 1 and 3 months after start of treatment, there was no significant difference compared to the before treatment values [79]. Due to poorly understood angiogenetic patterns of tumor microcirculation, biological agents such as VEGF inhibitors require further study.

Traditional chemotherapy with dacarbazine with addition of Selumetinib, a MEK1/2 inhibitor, has also recently been studied (NCT01974752) [80]. A total of 129 patients were enrolled, 97 receiving Selumetinib and dacarbazine and 32 receiving dacarbazine and placebo. In the dacarbazine/selumetinib group, the PFS was 2.8 months compared to 1.8 months in the placebo group. There was no difference in the OS between the two groups. The poor response of selumetinib prolonging the PFS or OS was attributed to adaptive resistance to MEK inhibition by tumor cells, which is supported in another studying using binimetinib, which showed similar poor response in the PFS [80].

### 8.1. Prospective Clinical Trials Comparing Treatment Efficacy of Multiple Approaches

Table 3 shows a review of clinical trials for the treatment of metastatic uveal melanoma. One trial involved an IV or hepatic artery infusion of 100 mg/m^2^ of fotemustine. Fotemustine belongs to the nitrosurea class of drugs. Fotemustine was administered on days 1, 8, 15 and 22 (HAI only) and every 3 weeks as maintenance. A total of 171 patients were enrolled in this trial. HAI did not increase survival compared to IV infusion, achieving an OS of 14.6 and 13.8 months, respectively [81]. HAI did produce a significant increase in the PFS compared to IV infusion, 4.5 and 3.5 months, respectively [81]. HAI also showed a greater radiographic response rate than IV infusion, 10.5% and 2.4%, respectively [81].

Another study enrolled 48 patients treated with HAI or IV fotemustine with a similar dosing schedule as the previous study. Patients were also treated with subcutaneous immune-modulating therapy with IL2 and TNF. Only one patient treated in this study showed CR and six patients showed PR [82]. A total of 18 patients showed SD [82]. Progressive free survival was approximately 1 year (range 273–810 days) [82]. In patients that did not show an PR or SD, the median OS was 321 days [82].

The addition of immune-modulating agents does not appear to provide any benefit when comparing the OS between the two trials where results were reported. The second study was completed (March 2008) and the authors concluded that the survival benefit of fotemustine with immune-modulating agents was better than non-fotemustine-treated patients [82]. The first study was ultimately terminated (June 2011) since this treatment did not increase the OS compared to other treatment options and delivery methods.

While patients predominately develop liver metastasis from uveal melanoma, extrahepatic metastasis is possible. Recent trials of widespread metastatic uveal melanoma such as PEMDAC trial investigated the use of pembrolizumab (PD1 inhibitor) and entinostat a class 1 HDAC inhibitor shown to promote immune checkpoint inhibition [83]. Immunotherapy, specifically targeting PD1 and CTLA4, has yet to show significant survival benefit. The authors suggest that epigenetic therapy with entinostat may promote PD1 inhibitors, increasing efficacy of immunotherapies. This trial is active but no longer recruiting participants. Another trial used ipilimumab and nivolumab and showed an OS of 12.7 months [84]. This trial was conducted in patients with widespread metastasis but showed that patients with exclusive liver metastasis had a worse OS compared to patients with extrahepatic metastasis, 9.2 and 23.5 months, respectively [84]. The authors concluded that patients with extrahepatic disease compared to exclusive liver metastasis showed increased benefit of combination therapy [84]. These data suggest that patients with liver only metastasis have even worse response and prognosis, highlighting the ongoing need for novel therapies in this patient population.

### 8.2. Persistent Need for Innovative Regional Therapies

Uveal melanoma is a devasting disease with a poor prognosis and limited survival. While leading therapies such as immune-modulating agents and checkpoint inhibitors have revolutionized the way we treat cutaneous melanoma, these methods are thus far not effective in uveal cases due to different molecular biology of uveal and cutaneous melanoma. This is why new drug discovery and innovative surgical approaches such as IHP or PHP that provide an aggressive regionalized approach continue to be necessary. Studies of IHP/PHP have suggested an OS up to 24 months [5]. While the median OS for IHP is 15.3 months, a single study has shown a median OS of 24 months. The OS results in this study were similar to surgical resection which had a median OS of 27 months [5]. A total of 65% of patients in this study had <10% liver involvement, therefore the low level of tumor burden for patients in this series may be a contributing factor to the prolonged median OS of 24 months compared to the median 15.3 month OS for IHP in the literature. This is an increase of 14 months survival compared to other treatments. A phase 3 clinical trial using IHP with melphalan will evaluate the effectiveness of this treatment by measuring the OS in 78 patients [85]. The trial will run from June 2013 to December 2021 and is recruiting patients (NCT01785316).

Current treatment of metastatic disease should focus on IHP due to failure of systemic chemotherapy, immunotherapy and other local regional approaches. Current alternative therapies for patients with liver metastases have proven to be less effective than IHP which has provided the longest median overall survival. However, the significant potential for morbidity and mortality is a clear limitation to this approach. Advances in surgical approaches such as PHP should focus on reducing the risk of systemic chemotherapy leak to make this approach more feasible. This will reduce risk of systemic toxicity, morbidity and limitations to this novel minimally invasive approach. Despite the relative improvement in survival with hepatic perfusion, novel therapy is desperately needed to improve outcomes for this disease. Newer approaches are investigating the effectiveness of combination therapies with targeted and immunotherapies in combination with local approaches such as TACE, Y-90 or ablation. Future directions should continue to evaluate the underlying biology of how the tumor cells evade the immune system and the development of novel therapies directed at these pathways. Current research is focused on dendritic cell vaccination and biospecific molecules such as tebentafusp. Tebentafusp in preliminary studies have shown to have the most promising effect compared to all other current immunotherapies. However, further research is needed to determine the effectiveness in overall survival. Molecular therapies with RNA vaccines focusing on the genomic mutations are currently being investigated in cases of cutaneous melanoma and have shown objective response in small studies [49]. These results may be extrapolated to cases of uveal melanoma offering personalized treatment based on the patient’s genetic mutations.

## 9. Conclusions

Uveal melanoma is a rare but very devastating disease with a poor long-term survival rate and over half of the patients developing metastasis. This cancer has a propensity to metastasize to the liver and many different techniques have been discovered to specifically treat it with modest outcomes. Unfortunately, immunotherapies have yet to be proven to be successful due to the many mechanisms that this type of cancer has developed. Additional research and clinical trials are needed to evaluate the utility of future targeted small-molecule inhibitors and how the body evades the immune system to develop better immunotherapies for the treatment of uveal melanoma. Currently, the most effective therapy for liver metastases from uveal melanoma continues to be isolated hepatic perfusion.

## Figures and Tables

**Figure 1 cancers-13-03426-f001:**
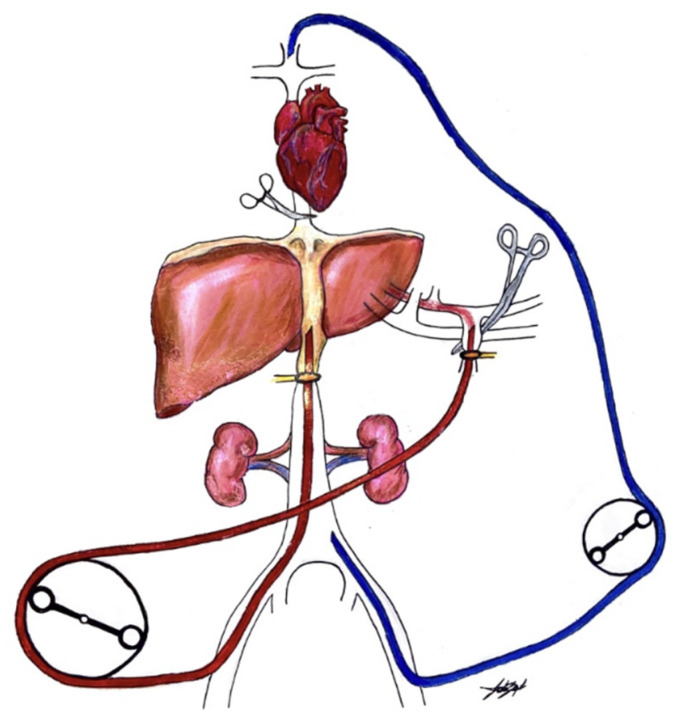
Venovenous bypass circuit.

**Table 1 cancers-13-03426-t001:** Review of literature, open isolated hepatic perfusion. CR = complete response, PR = partial response, SD = stable disease, and PD = progressive disease.

Author	Year	N	Disease	Perfusant	OS (mo)	CR	PR	SD	PD
Ben-Shaat et al. [36]	2017	52	Uveal	Melphalan buffered (*n* = 36) Melphalan without buffer (*n* = 16)	Buffer, 24.2 Without buffer, 26	Buffer 11% Without buffer 44%	Buffer 47% Without buffer 44%	Buffer 20% Without buffer 12%	Buffer 22% Without buffer 0%
de Leede et. al. [37]	2016	30	Uveal	Melphalan/Oxaliplatin	10	-	-	-	-
Ben-Shabat et al. [34]	2016	68	Uveal	Melphalan/Oxaliplatin/TNF-alp	22.4	20%	48%	20%	12%
van Iersel et. al. [38]	2014	11	Colorectal (*n* = 8) Uveal (*n* = 3)	Melphalan	18.7	-	*n* = 3, 1 OM, 2 CRC	-	*n* = 5, 2 OM, 3 CRC
Olofsson et. al. [5]	2014	34	Uveal	Melphalan	24	12%	56%	18%	15%
Varghese et al. [35]	2010	17	Uveal	Melphalan	11.9	*n* = 1	*n* = 7	-	-
van Etten et al. [39]	2009	8	Uveal	Melphalan	11	-	37.50%	37.50%	25%
Rizell et al. [40]	2008	27	Occular (20)Cutaneous (5) Anal (2)	Melphalan/TNF/Cisplatin (11) Melphalan (16)	12.6	7%	63%	7%	-
van Iersel et al. [41]	2008	12	Uveal	Melphalan	10	0%	33%	50%	-
Noter et al. [42]	2003	8	Uveal	Melphalan	9.9	0%	50%	25%	-
Alexander et al. [43]	2003	29	Uveal	Melphalan	12.1	10%	52%	-	-
Alexander et al. [33]	2000	22	Uveal	Melphalan (11) Melphalan/TNF (11)	11	9.50%	52%	-	-

**Table 2 cancers-13-03426-t002:** Review of literature, percutaneous isolated hepatic perfusion.

Author	Year	N	Disease	Perfusant	OS (mo)	CR	PR	SD	PD
Artzner et. al. [46]	2019	16	Uveal	Melphalan	27.4	-	First PHP 60% Second PHP 67%	First PHP 33% Second PHP 33%	7%
Karydis et. al. [44]	2017	51	Uveal	Melphalan	15.3	5.90%	43.10%	33.30%	-
Vogl et al. [47]	2017	18	Uveal	Melphalan	9.6		First PHP 44% Second PHP 89% Third PHP 83%	First PHP 39% Third PHP 17%	First PHP 17% Second PHP 11%
Hughes et al. [45]	2016	93	Uveal (*n* = 83) Cutaneous (*n* = 10)	Melphalan (*n* = 44) BAC (*n* = 49) systemic chemo (*n* = 24), 49% chemoembolization (*n* = 11), 22.4% radioembolization (*n* = 3), 6.1% systemic chemo/embolization (*n* = 1), 2% surgery (*n* = 1), 2% supportive care (*n* = 9), 18.4%	PHP 10.6 BAC 10.0	-	PHP 36.4% BAC 2%	PHP 52.3% BAC 40.8%	-
Forster et al. [48]	2014	10	Uveal (*n* = 5) Cutaneous (*n* = 3) Unknown (*n* = 1) Sarcoma (*n* = 1)	Melphalan	24.2	-	SD or PR 90%	-	-

**Table 3 cancers-13-03426-t003:** Review of clinical trials for uveal melanoma with liver metastasis.

Status	Identifier	Intervention	Phase	OS	PFS	RR
Terminated	NCT00110123	Drug: Fotemustine	Phase 3	HAI: 14.6 months	HAI: 4.5 months	HAI: 10.5%
		Drug: HIA		IV: 13.8 months	IV: 3.5 months	IV: 2.4%
No longer available	NCT01728051	Drug: Melphalan	-	PHP-Mel: 10.6 months	PHP-Mel: 5.4 months	PHP-Mel: 23.7%
		Device: Percutaneous Hepatic Perfusion		BAC: 10 months	BAC: 1.6 months	BAC: 4.1%
Completed	NCT00062933	Procedure: Laparotomy	Phase 2	OS for patient with OR 581 days (346–826)	155–855 days, median ~1 year	
		Drug: Fotemustine				
		Drug: Interferon Alpha		OS for patients with SD 448 days (175–1020)		
		Drug: Interleukin 2				
				Not significantly improved		
				OS in patients with neither OR or SD was 321 days		
Recruiting	NCT02913417	Device: SIR-Spheres Yttrium 90	Phase 1	-	-	-
		Drug: Ipilimumab	Phase 2			
		Drug: Nivolumab				
Completed	NCT00324727	Drug: Melphalan	Phase 3	10.6 months PHP-Mel	5.4 months PHP-Mel	27.3% PHP-Mel
		Drug: Regional Chemotherapy				
		Drug: Systemic Chemotherapy		10.0 months BAC	1.6 months BAC	4.1% BAC
		Procedure: Hepatic Artery Embolization				
Terminated	NCT01730157	Biological: Ipilimumab	Early Phase 1	-	-	-
		Radiation: Yttrium Y 90 Glass Microspheres				
		Other: Laboratory Biomarker Analysis				
Completed	NCT01311466	Procedure: Liver Transplantation	Not Applicable	-	-	-
Active, not recruiting	NCT01473004	Device: Sir-Spheres^®^	Phase 2	-	-	-
Completed	NCT03408587	Biological: CVA21	Phase 1	-	-	-
		Biological: Ipilimumab				
Recruiting	NCT02936388	Procedure: SIRT	Phase 2	-	-	-
		Procedure: DSM-TACE				
Recruiting	NCT01785316	Procedure: IHP	Phase 3	-	-	-
Not yet recruiting	NCT04184518	Drug: Cediranib Maleate	Phase 2	-	-	-
		Drug: Durvalumab				
Not yet recruiting	NCT04728633	Drug: Carmustine Drug: Ethiodized Oil Procedure: Transarterial Chemoembolization Other: Medical Device Usage and Evaluation	Phase 2	-	-	-
Recruiting	NCT04463368	Procedure: Isolated Hepatic Perfusion Drug: Ipilimumab Drug: Nivolumab	Phase 1	-	-	-
Completed	NCT00661622	Drug: GM-CSF Procedure: Embolization	Phase 2	Immunoembolization 21.5 months Plain embolization 17.2 months	Immunoembolization 3.9 months Plain embolization 5.9 months	Immunoembolization 21.2% Plain embolization 16.7%
Not yet recruiting	NCT04812470	Drug: Autologous Tumor-Infiltrating Lymphocytes Drug: Melphalan Drug: Interleukin-2	Phase 1	-	-	-
Recruiting	NCT03068624	Biological: Aldesleukin Biological: Autologous CD8+ SLC45A2-specific T Lymphocytes Drug: Cyclophosphamide Biological: Ipilimumab	Phase 1	-	-	-
Completed	2008-001974-33	mRNA-Transfected Dendritic Cell Vaccination	-	-	-	74% of patients demonstrated presence of tumor-specific T cell at biopsy after vaccination
Completed	2010-022687-12	Sorafenib	Phase 2	-	-	-
Ongoing	2011-004200-38	Ipilimumab with Radiofrequency Ablation	Phase 2	-	-	-
Ongoing	2014-002439-32	Tranarterial Radioembolization with Y-90 vs. TACE with Cisplatin	Phase 2	-	-	-
Ongoing	2020-003188-24	IHP with Ipilimumab and Nivolumab	Phase 1	-	-	-
Ongoing	2019-001045-40	Durvalumab with Cediranib	Phase 2	-	-	-
Completed	2010-021946-22	Ipilimumab	Phase 2	-	-	-
Ongoing	2019-000657-31	Adoptive Therapy with TCR Gene Engineered T Cells	-			
Completed	2010-023058-35	Autologous TriMix-DC Therapeutic Vaccine in Combination with Ipilimumab	Phase 2	-	51% at 6 months	38% RR 8/38 patient CR 7/38 patient PR
Ongoing	2015-000417-44	Melphalan/HDS	-	-	-	-

## Data Availability

Not applicable.

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
