# Peer review of "Poor Response to Checkpoint Immunotherapy in Uveal Melanoma Highlights the Persistent Need for Innovative Regional Therapy Approaches to Manage Liver Metastases"

_cancers, 2021, doi:10.3390/cancers13143426_

Round 1

Reviewer 1 Report

Dear authors,

In general, I like the article and with so many evolving anti-cancer treatments it is good to have an update on trials and treatments of metastasized UM.

Unfortunately, the introduction on uveal melanoma is insufficient, and sometimes incorrect, i.e. the metastases rate in UM patients. Surprisingly, in te conclusion you do give the right percentage of UM patients with metastases.

Also the hematogenous route of dissemination is not mentioned!

It is clear non of the authors is an ophthalmologist, however information is broadly available, for instance read Uveal melanoma, review 2020 (Jager et al).

It would be good to elaborate on the genetic background of UM as well., because herein might be a solution for future targeted therapies.

Probably in the introduction, and otherwise in the part on molecular targeted therapies. For instance read Uveal melanoma: Towards a molecular understanding. (Smit et al)

Furthermore, I think the authors should be aware of trials conducted outside the US, especially if they attempt to review all available treatments of UM liver metastases.

Specific Comments:

Page 2.

Line 44. Add posterior uveal tract of the eye.

Line 45. 90% of cases > rephrase. 90% of UM metastazis tot he liver. Up to 50% of UM will develop metasteses.

Primary Treatment for Localized Uveal Melanoma

radiation techniques : elaborate on the commonly used treatments: protonbeam and fractionated stereotactic irradiation.

I miss endoresection of the primary tumor as surigcal technique (Damato, Kilic)

Line 86 en 87

Uveal melanoma can spread via conjunctival lymphatics to other regions of the body and metastasize in other organs.

The metastasis spread hematogeneous and not via the lymphatics.

 In cases of uveal melanoma, 5 and 10 year rate of metastasis is 25% and 34% respectfully [1]

>The rates of metastasis are much higher. Up to 50%

Page 3. Line 135

Surgical resection is the preferred treatment for patients who are medically fit for surgery and are resectable> patients are not resectable, maybe the metastasis are?

Pqge 5/6 Surgical Approaches and Treatment Outcomes for Isolated Hepatic Perfusion

I miss comments on morbidity and hospital admission due tue treatment.

Part of the overal survival is not necessarily beneficial for the patient because these are days patient is ill and hospitalized.

-Can you comment on morbidity and duration of surgery of IPH compared to PHP

-Are there any papers on IPH compared tot PHP?

Page 9. Exploration into Immunotherapy for Treatment of Metastatic Uveal Melanoma

This chapter is missing all sorts of immuntherapies for metastatic UM.

What about dendritic cell therapy? Several studies in literature are available (De Vries et al). Even a review on Immunotherapies for the Treatment of Uveal Melanoma-History and Future. is available.

Page 11. Line 402 immunoprivilege> immune privilege

Page 11. Advances in Molecular Targeted Therapies in Uveal Melanoma

>There is a difference in genes involved in the development of UM and the development of UM metastasis. Therefor, other genes need to be targeted in treatment of |UM metastesis. See the mentioned review previously.

Page 12. Line 443 Increase> increased

Page 12/13/14 Table 3. Review of clinical trials for uveal melanoma with liver metastasis.

These are only US based studies. Not European studies. This should be explicited.

Page 14 line 491

Studies of IHP/PHP have suggested an OS up to 26 months [3]. This is an increase of 14 months survival compared to other treatments.

>This is just one study showing such prolonged survival. None of the others have such OS with PHP. Treated number of patients in this study are 16. How come this study is the only one with patients with such long OS? Please comment.

Reviewer 2 Report

The review article by Szeligo et al. undertook an effort in highlighting recent developments in the treatment options for liver metastases from uveal melanoma, with a particular focus on isolated hepatic perfusion (IHP). Indeed the IHP is an optimal therapeutically approach that allows high dose chemotherapy with melphalan to be delivered directly to the liver without systemic effects. The manuscript is comprehensive to the topic and meticulously written. The strength of this manuscript is an excellent chronological order to the matter covered and self-explanatory tables.

The reviewer has the following suggestion.

  1. The reviewer believes that there should be a chapter discussing the limitation of IHP to manage liver metastases from uveal melanoma. A given approach can't be perfect, and there always exists a restriction. 
  2. In addition to IHP, a chapter on the future direction or alternative therapy is missing. Adding a chapter on this topic shall give additional value to this review article. 

Round 2

Reviewer 2 Report

Thanks for answering all comments. 

Author Response

no further comments